


# High-resolution Emission Inventory Development and Co-emission Hotspot Identification of Air Pollutants and Greenhouse Gases in Central Plains Region, China

Jie Li<sup>1,2</sup>, Shasha Yin<sup>1,2</sup>, Conghui Su<sup>1,2</sup>, Jiaxin Wei<sup>1,2</sup>, Mingyue Guan<sup>1,2</sup>, Chong Yu<sup>2,3</sup>

<sup>1</sup>School of Ecology and Environment, Zhengzhou University, Zhengzhou, 450000, China
 <sup>2</sup>Research Institute of Environmental Sciences, Zhengzhou University, Zhengzhou, 450000, China
 <sup>3</sup>College of Chemistry, Zhengzhou University, Zhengzhou, 450000, China

Correspondence to: Shasha Yin (E-mail: shashayin@zzu.edu.cn)

Abstract. A high-resolution inventory provides scientific basis for numerical simulations and control strategies. Under the background of synergistic carbon reduction and pollution control, constructing a carbon-pollutant co-emission inventory is of great significance for regional air quality improvement. Taken Henan Province in the Central Plains region as an example, the most polluted regions in China, an update emission inventory was developed. The study presents results showing that in 2022, the total emissions of SO<sub>2</sub>, NO<sub>X</sub>, CO, PM<sub>10</sub>, PM<sub>2.5</sub>, VOCs, NH<sub>3</sub>, BC, OC, CO<sub>2</sub>, CH<sub>4</sub>, and N<sub>2</sub>O in Central China, particularly Henan Province, were 408.7, 1336.2, 4647.3, 901.1, 440.0, 759.3, 672.7, 47.4, 90.3, 540462.0, 12462.0 and 42.9 kt respectively. The emissions were predominantly attributed to industrial combustion, electricity generation, motor vehicles, and agricultural activities. Significant spatial heterogeneity was observed. Northern heavy industrial cities exhibited high carbon and pollution intensities with carbon emission 1.75-3.7 times higher than the provincial average. In contrast, central transportation hubs were primarily characterized by elevated emissions of NO<sub>X</sub> and VOCs. Southern agricultural areas showed low carbon but high NH3 emissions. Temporally, emissions of SO2 and PM2.5 peaked during winter, whereas NH3 increased during the summer agricultural season. High-emission grids were predominantly concentrated in urban agglomerations of the north-central region, especially around Zhengzhou, Jiaozuo, and Anyang. Hotspot analysis revealed that 5% of high-emission grids accounted for more than 50% of total emissions, indicating a highly uneven spatial distribution. These results highlight that understanding the region-specific emission characteristics of different regions is critical for developing mitigation strategies in future.

### 1 Introduction






Atmospheric pollutants are primarily emitted due to fundamental human activities, including energy consumption and economic operations (Kanakidou, 2024; Huang et al., 2024). Notably, pollutants such as SO<sub>2</sub>, NO<sub>X</sub>, VOCs, PM<sub>2.5</sub> and NH<sub>3</sub>, play a key role in regional atmospheric pollution formation (Kanakidou, 2024), which can significantly impair visibility, ecology and public health (Manisalidis et al., 2020). Meanwhile, the global warming effects of greenhouse gases (GHGs), as the main emission drivers of climate change, have been widely noticed and recognized (IPCC, 2021). Although air pollutants and GHGs differ in their environmental impacts with air pollutants primarily posing a local, short-term health and ecological risk, while GHGs contribute to global-scale, long-term warming effect, they are fundamentally both gaseous originating from highly overlapping emission sources, including fossil energy combustion, industrial process, transportation, and residential activities (Xu et al., 2025; Liu et al., 2021; MEEPRC, 2021).

Globally, the air pollution situation in China remains concerning and the PM<sub>2.5</sub> and ozone pollution have persisted at high concentrations for a long time (IQAIR, 2024; Jin et al., 2021). According to the Bulletin of China's Ecological and Environmental Conditions 2023, 136 out of 339 cities at prefecture level and above in the country have exceeded national Grade II air quality standards (MEEPRC, 2024a; Wang et al., 2025). Meanwhile, China is a major contributor to global GHG emissions. According to the Carbon Emissions Report 2023 released by International Energy Agency, the total carbon emissions in China reach 12.6 billion tons in 2023 with a growth rate of 4.7% (IEA, 2024). Therefore, more and more scholars at home and abroad are highly concerned about the quantification of their emissions including Chinese scholars as well.

Domestic and international researches have made significant progress, and a series of national and global scale emission inventories have been constructed, including MEIC (Li et al., 2018), EDGAR (Muntean et al., 2018), REAS (Kurokawa et al., 2013), and CHRED (Cai et al., 2018). Meanwhile, a series of emission inventories in China have been constructed at the regional (Xu et al., 2020; Zhao et al., 2024; Wu et al., 2022), provincial (Zhou et al., 2020; Jiang et al., 2020a), urban (Liu et al., 2018), and industry (Gao et al., 2019; Hua et al., 2016; Jiang et al., 2020b) levels have also made significant progress. In particular, the development of air pollutant emission inventories has been rapid





advancements in spatial resolution and temporal distribution (Chen et al., 2024), and some studies have successfully constructed gridded data with a precision of  $0.1^{\circ} \times 0.1^{\circ}$  or even higher (Zheng et al., 2021). However, GHGs emission inventories typically emphasize aggregate total emission of the five major sectors, including energy, industrial process, agriculture, forestry, and waste at national or provincial levels, rather than high-resolution emission inventories (Geng et al., 2011; Yu and Tan, 2023; Xue et al., 2021; He et al., 2024). At present, China has attached great importance to the work of reducing pollution and carbon emissions in a coordinated manner as a national strategy (CPGPRC, 2022; Yu et al., 2020). A series of policy documents have been issued successively which cover the national or provincial strategy (CPGRC, 2024a) and key industries (CPGRC, 2024b; MTPRC, 2024). As the emission reduction potential of traditional end-of-pipe measures gradually narrowing (Kong et al., 2024; Cao et al., 2025), it is necessary to develop integrated emission inventory of carbon and pollutants at city level or even enterprise level (Wu et al., 2024). It not only aids to explore the emission reduction potential from the front-end and also enhances the identification of GHG spatial grid synergies.

In the dual context of high-value for air pollutant and GHG emissions pressure in China, Central China, especially represented by Henan Province is typical and representative of high pollutant concentrations and high carbon emission. Henan Province is a major contributor to PM<sub>2.5</sub> and O<sub>3</sub> pollution double high value area: in 2023, PM<sub>2.5</sub> concentration exceeds the national standard by 28%, and O<sub>3</sub> exceeds the national standard by 12%; the proportion of good air quality day was lower than the national average of 85.5% (MEEPRC, 2024a). Also, Henan Province has a share of coal in energy consumption close to 55.3% in 2023, and the energy-related CO<sub>2</sub> emission in this region was accounted for approximately 5% of the total emission in China (HBS, 2024). Therefore, we selected Henan Province as the study area to develop a integrated emission inventory of carbon and pollutants based on a unified source classification system with enterprise-level point source support as possible. It hopes to features enterprise-level point source support, 3 km × 3 km spatial grid resolution, and detailed temporal allocation.

The objectives of this study are to (1) calculate the emissions of air pollutants and GHGs based on uniform sources, including nine conventional air pollutants (SO<sub>2</sub>, NO<sub>X</sub>, CO, PM<sub>10</sub>, PM<sub>2.5</sub>, VOCs, NH<sub>3</sub>, BC, and OC) and three GHGs (CO<sub>2</sub>, CH<sub>4</sub>, and N<sub>2</sub>O); (2) analyze the emission characteristics of source






and city contributions; (3) construct a 3 km × 3 km spatially gridded emissions and monthly assignments; (4) compare with other studies and conduct uncertainty analysis; and (5) identify high-emission grids using hotspot analysis methods.

#### 2 Data and methodology

#### 2.1 Study domain

This study centered on Henan Province as its primary research site, with 2022 serving as the reference period. Situated at the crossroads of central and eastern China, the region stretches along the middle and lower reaches of the legendary Yellow River, occupying a strategically important position in the country's geographical landscape. It governs 18 administrative divisions, including Zhengzhou, Kaifeng, Luoyang, Pingdingshan, Anyang, Hebi, Xinxiang, Jiaozuo, Puyang, Xuchang, Luohe, Sanmenxia, Nanyang, Shangqiu, Xinyang, Zhoukou, Zhumadian, and the Jiyuan Demonstration Area (see Figure 1).

Henan Province, a central Chinese region, exemplifies high population density, industrial and agricultural prominence, and key transportation networks. With a landmass of roughly 167000 square kilometers, Henan Province makes up about 1.73% of China's total territory (HPPG, 2023). In 2022, its GDP reached 6.13 trillion yuan, ranking fifth nationally (HBS, 2023; SCPRC, 2023). With a permanent population of 98.72 million in 2022, it ranks third in China (NBS, 2023). The economic landscape is dominated by a split of 8.6% in the primary sector, 38.3% in the secondary, and 53.1% in the tertiary sector (HBS, 2023). The region's energy structure has long been dominated by coal, with standard coal consumption reaching 214 million tons in 2022, accounting for 62.7% of the province's total energy consumption (HBS, 2023). The five major high-energy-consuming industries of petrochemicals, chemicals, building materials, iron and steel, and non-ferrous metals account for 53.84% of the total output value of large-scale industrial products in Henan Province (HBS, 2023). Henan had 42.6 million live pigs in stock in 2022, ranking second nationally, while Zhengzhou, the provincial capital, had 4.57 million registered motor vehicles, ranking sixth among Chinese cities (HBS, 2023; HPDEE, 2022).

### 110 Figure 1. Study domain.



# 2.2 Source categorization

To develop a trustworthy emissions inventory, as a fundamental requirement, it's vital to have a sound and thorough system for classifying the emission sources (Zhang et al.,2023). Distinct emission inventories traditionally cataloged GHGs and air pollutants separately. To facilitate the integration of the compilation processes for both types of emission inventory, this study focuses on the unified source classification framework. By refer to the "Technical Guidelines for the Compilation of Comprehensive Emission Inventories of Air Pollutants and GHGs (for Trial Implementation)" (hereinafter referred to as the "Technical Guidelines") (MEEPRC,2024b), the unified classification framework of seven emission source of electricity and thermal power, industry, mobile source and oil storage and transportation, fugitive dust, resident, agriculture, and waste treatment was adopted. A detailed list of these specific



source types and associated activity data can be found in Table S1.

#### 2.3 Methodology and data collection

The research utilized two main techniques for gauging emissions: the emission factor and the mass balance methodologies. Data on activities were gathered from a blend of on-site investigations, documented information from ecological and environmental authorities, and significant statistical compilations, including the Henan Provincial Statistical Yearbook, the China Statistical Yearbook, and the China Agricultural Yearbook. The compiled datasets included pivotal elements such as energy usage, industrial output, pollution abatement facilities, regional dispersion, and vehicle numbers, with comprehensive data outlined in Table S1.

Emission factors were chosen using pertinent literature and local assessments, emphasizing applicability to Henan Province and adjacent regions. The corresponding factor data can be found in Tables S2 through S7.

# 2.3.1 Air pollutant emission calculation

The emissions of conventional air pollutants ( $SO_2$ ,  $NO_X$ , CO, VOCs,  $PM_{10}$ ,  $PM_{2.5}$ ,  $NH_3$ ) are calculated using the emission factor method with the following formula:

$$E=A\times EF$$
 (1)

Here, E denotes the total emissions from each source within Henan Province, A represents the activity level data, EF stands for the emission factor value.

For electricity and thermal power source as well as industrial combustion, a more detailed mass balance method can be used for SO<sub>2</sub> emission, with the following formula:

$$E_{SO_2} = \sum_{p,q} 2 \times S \times F_{p,q} \times C_q \times \left(1 - \eta_{SO_2}\right) (2)$$

where, E is the emission of SO<sub>2</sub>, 2 is the molecular weight ratio of sulfur dioxide (SO<sub>2</sub> molecular weight 64) to sulfur (S molecular weight 32), S is the sulfur content in the fuel, F is the fuel consumption, C is the sulfur conversion rate, and  $\eta_{SO_2}$  is the sulfur removal rate.

BC and OC emissions are derived from PM<sub>2.5</sub> levels and their respective fractions, calculated as follows:

$$E_i = E_{PM_{2.5}} \times f_i$$
 (3)

Where  $E_i$  is the emission of BC or OC; i represents BC or OC;  $E_{PM_{2.5}}$  is the PM<sub>2.5</sub> emission; and  $f_i$  denotes the BC or OC fraction within PM<sub>2.5</sub>.

#### 150 2.3.2 GHG emission calculation

(1) CO<sub>2</sub> emission calculation method

CO<sub>2</sub> emissions from electricity and thermal power source are calculated using the following formula:

$$E_{CO_2} = \sum_i AC_i \times NCV_i \times CC_i \times COF_i \times \frac{44}{12}$$
 (4)

where,  $E_{CO_2}$  is the CO<sub>2</sub> emission; i is the fuel type; AC is the fuel consumption; NCV is the net calorific value of the fuel; CC is the carbon content per unit calorific value of the fuel; COF is the carbon oxidation factor of the fuel; and 44/12 is the molecular weight ratio of CO<sub>2</sub> to C.

The CO<sub>2</sub> emission calculation methods for industrial source, mobile source and oil storage and transportation, residential source, and agricultural source are similar to those for electricity and thermal power source, all based on fuel consumption and emission factors.

(2) CH<sub>4</sub> emission calculation method

The CH<sub>4</sub> emission calculation formula for agricultural livestock and poultry manure management systems is:

$$E_{M-CH_4}=EF_{M-CH_4}\times AP$$
 (5)

where,  $E_{M-CH_4}$  is the CH<sub>4</sub> emission factor per unit livestock, and AP is the annual livestock population.

For waste treatment source, the CH<sub>4</sub> emission formula for domestic wastewater treatment is:

$$E_{CH_4}$$
=(TOW×EF)-R (6)

where, TOW is the total organic loading in domestic wastewater, EF is the emission factor, and R is the CH<sub>4</sub> recovery rate.

The CH<sub>4</sub> emission formula for municipal solid waste landfill is:


$$E_{CH_4} = (MSW_T \times MSW_F \times L_0 - R) \times (1 - OX)$$
 (7)

where,  $MSW_T$  is the total municipal solid waste generation,  $MSW_F$  is the fraction sent to landfills,  $L_0$  is the methane generation potential, R is the recovered methane, and OX is the oxidation factor.

CH<sub>4</sub> emissions from mobile source and oil storage and transportation and residential source are calculated using Eq. (1)

#### (3) N<sub>2</sub>O Emission Calculation Method

 $N_2O$  emissions from agricultural livestock and poultry farms are divided into direct and indirect emissions. The direct emission formula is:

$$E_{M-N_2O-dir} = \sum_i AP \times Nex \times \frac{MS_i}{100} \times EF_i \times \frac{44}{28}$$
 (8)

where, Nex is the annual nitrogen excretion per animal unit,  $MS_i$  is the fraction of manure managed using system i,  $EF_i$  is the direct N<sub>2</sub>O emission factor for management system i, and AP represents livestock population (using annual throughput for pigs, broilers, and beef cattle, and average inventory for dairy cattle and laying hens).

The indirect emission formula is:

$$E_{\text{M-N}_2\text{O-ind}} = \sum_{i} \text{Nex} \times \text{MS}_i \times \frac{\text{Frac}_{gas}}{100} \times \text{EF}_1 \times \frac{44}{28} \times \text{AP} \times 10^{-3}$$
 (9)

where  $Frac_{gas}$  is the fraction of nitrogen volatilized as NH<sub>3</sub> and NO<sub>X</sub>, and  $EF_I$  is the emission factor for N<sub>2</sub>O formation from volatilized nitrogen.

The  $N_2O$  emission calculation formula for domestic wastewater treatment in waste treatment source is:

$$E_{N_2O} = N_E \times EF_E \times \frac{44}{28}$$
 (10)

where  $N_E$  is the nitrogen content in wastewater,  $EF_E$  is the N<sub>2</sub>O emission factor for wastewater treatment, and 44/28 is the molecular weight conversion factor.

 $N_2O$  emissions from electricity and thermal power and mobile source and oil storage and transportation are calculated using Eq. (1)

# 195 2.4 Temporal and spatial allocation


This research aimed to bolster the precision of air quality modeling at a high resolution, and to gain a deeper understanding of how air pollutants and greenhouse gases are released together. To achieve this, we crafted a detailed gridded emission inventory that captured both time-based and spatial-based nuances. Based on annual emissions, the results were further allocated by month and mapped to a  $3 \text{ km} \times 3 \text{ km}$  spatial grid.

For temporal distribution, monthly variation was determined based on activity profiles of different emission source. Key monthly activity parameters were collected through sector-specific surveys and







field investigations. These parameters included electricity generation, traffic volume, satellite fire counts, residential activity patterns, construction schedules, and precipitation. Using these data, weighted monthly coefficients were developed to reflect the temporal variation of different source. These coefficients were then applied to allocate annual emissions across individual month, resulting in a time-resolved emission profile.

Although high-resolution spatial allocation methods for air pollutants are relatively well established, grid-based mapping of GHG emissions remains limited. In particular, refined spatial distribution of GHGs has lacked systematic methodological approaches. In this study, spatial allocation was conducted using ArcGIS software, with emissions mapped to a 3 km × 3 km grid. The allocation process emphasized enhancing the representation of point-source. For major sectors such as power, steel, cement, and chemicals, geographic coordinates of 7390 enterprises were collected. These covered 23 key industries, including cement, glass, and iron and steel. Line source were allocated based on road networks and traffic volume, while area source were distributed using land use and population density data. The final gridded inventory demonstrates high spatial resolution and a high coverage rate of point source. The uncertainty of the grid spatial distribution is mainly due to the deviation of proxy data such as population density and cultivated land area in spatial distribution, as well as the inaccuracy of the geographical coordinates of pollution sources. Detailed temporal and spatial allocation parameters are provided in Tables S8–S9.

#### 2.5 Identification method of hotspot regions for co-control

To pinpoint areas where CO<sub>2</sub> and key air pollutants overlap, this research leveraged a hotspot analysis. We employed ArcGIS to conduct and visualize this analysis. Initially, using our pre-existing 3 km × 3 km gridded emission inventory, we mapped out the spatial distribution of CO<sub>2</sub> alongside four pollutants of target—namely SO<sub>2</sub>, NOx, PM<sub>2.5</sub>, and VOCs—across the entire grid network of Henan Province. Each pollutant, including CO<sub>2</sub>, was ranked independently according to grid-level emission intensity. Grids within the top 20% were classified into four hotspot categories: the top 5% as Category 1, 6%–10% as Category 2, 11%–15% as Category 3, and 16%–20% as Category 4. A bi-directional matching rule was then applied to identify co-hotspots. For example, if a grid ranked within the top 20% for both CO<sub>2</sub> and SO<sub>2</sub> emissions, it was identified as a CO<sub>2</sub>–SO<sub>2</sub> co-hotspot. The same method was used to identify co-hotspot zones for CO<sub>2</sub> with NO<sub>x</sub>, PM<sub>2.5</sub>, and VOCs. These results provide spatial insights






into areas with high levels of both carbon and pollutant emissions.

#### 3 Results and discussions

#### 3.1 Emissions and source contribution

As shown in Table 1 and Figure 2, the data illustrated the composition of atmospheric pollutants and GHGs emitted from the seven primary source categories in Henan Province. In 2022, the cumulative emissions were staggering, with SO<sub>2</sub> contributing 408.7 kt, NO<sub>x</sub> at 1336.2 kt, CO at 4647.3 kt, PM<sub>10</sub> at 901.1 kt, PM<sub>2.5</sub> at 440.0 kt, VOCs at 759.3 kt, NH<sub>3</sub> at 672.7 kt, BC at 47.4 kt, OC at 90.3 kt, CO<sub>2</sub> at 540462.0 kt, CH<sub>4</sub> at 12462.0 kt, and N<sub>2</sub>O at 42.9 kt.

SO<sub>2</sub> emissions are mainly from industrial source, accounting for 83.5% of the total emission. Among the industrial source, industrial combustion were the most dominant sub-category, accounting for 47.2%, followed by cement manufacturing (12.1%) and chemical raw material manufacturing (11.6%). Electricity and thermal power constituted the second most important source category, accounting for 5.0% of the total. The internal emission from this sector was almost exclusively attributable to coal combustion, which contributed 99.5% of the emissions. This is closely related to Henan's coal-dominated energy structure. In 2022, coal accounted for 62.7% of the province's total energy consumption, significantly higher than the national average (HBS, 2023). The extensive use of coal and high-sulfur fuels has led to the industrial and power sectors being the major emission source of SO<sub>2</sub>, a conclusion that aligns with the results of previous studies (Tong et al., 2025; Ren et al., 2021).

NO<sub>X</sub> emissions are primarily attributed to mobile source and oil storage and transportation, industrial source, and electricity and thermal power, account for 39.8%, 36.2%, and 21.9% of the total emission, respectively. Within the mobile source and oil storage and transportation, On-road mobile dominates significantly, contributing to 31.1% of the total emissions. As a national transportation hub, Henan has experienced increasing road traffic pressure. By 2022, the total length of urban roads reached 19532 km, 1.81 times that of 2012 (HBS, 2013; HBS, 2023). During the same period, the number of heavy-duty trucks increased by approximately 1.52 times (HBS, 2013; HBS, 2023). The intensive operation of diesel trucks, especially in logistics centers such as Zhengzhou and Luoyang, has significantly increased NO<sub>X</sub> emissions. Within industrial sources, industrial combustion (64.2%)






dominated the cement industry. Same to SO<sub>2</sub>, within electric thermal sources, coal combustion contributed up to 99.2% of NO<sub>X</sub> emission (Sun et al., 2023).

CO emissions are dominated by industrial sources (57.1%), followed by mobile source and oil storage and transportation (16.9%) and residential sources (16.7%). Among the subcategories of industrial source, industrial combustion accounts for 19.1%, ferrous metal production for 17.3%, and cement manufacturing for 9.5%. Within the mobile source and oil storage and transportation category, on-road mobile is the primary contributor, accounting for 93.1% of the total emission from this sector (He et al., 2024). The increasing number of heavy-duty diesel trucks has intensified transportation activity, thereby significantly contributing to CO emissions (Keramydas et al., 2019). Residential CO emissions mainly originate from firewood use (7.6%) and household coal combustion (6.5%), indicating that traditional energy use in rural and peri-urban areas remains a key emission source. The transition to clean energy alternatives in these regions remains insufficient and requires further promotion.

PM<sub>10</sub> and PM<sub>2.5</sub> were mainly from fugitive dust, industrial, and residential. Fugitive dust source contributed 49.0% of PM<sub>10</sub>, while industrial source accounted for 21.2%. Within sub-sources of emissions from dust source, road dust (23.0%) and construction dust (25.9%) were predominant, showing that significant contributions of transportation and urban construction to particulate emissions. From 2012 to 2022, the length of urban roads in this region increased by 8734 km (an 81.0% rise), and the total area under building construction grew by 64.4% (HBS, 2013; HBS, 2023). Urban expansion and infrastructure development have significantly increased dust emissions from roads and construction sites (Zhou et al., 2023). On the other hand, PM<sub>2.5</sub> was mainly derived from industrial source, dust source and residential, accounting for 43.7%, 24.4% and 18.0%, respectively. In the residential sector, biomass combustion contributes to 5.6% of PM<sub>2.5</sub> emissions, primarily resulting from the use of firewood and crop residues in rural areas. This indicates that the adoption of clean energy alternatives remains insufficient, positioning biomass combustion as a significant contributor.

The primary sources of VOCs emissions are predominantly industrial, residential, and mobile source and oil storage and transportation source, accounting for 38.3%, 32.3%, and 24.1% of the total emissions, respectively. Within sub-sources of emissions from industrial source, chemical manufacturing (Zhou et al., 2021), non-ferrous metals, and industrial combustion together account for 14.0% of VOCs emissions. This reflects the dominance of traditional heavy industries in Henan. The relatively high

Table 1. Emissions of air pollutants and GHGs in Henan Province by source category, 2022 (Unit: kt).

| Emission source                                        | $SO_2$ | NOx    | 00     | $PM_{10}$ | PM <sub>2.5</sub> | VOCs  | NH <sub>3</sub> | BC   | 00   | CO <sub>2</sub> | CH4     | N <sub>2</sub> O |
|--------------------------------------------------------|--------|--------|--------|-----------|-------------------|-------|-----------------|------|------|-----------------|---------|------------------|
| Electricity and thermal power                          | 20.5   | 292.6  | 264.0  | 39.1      | 22.5              | 5.3   | 2.9             | 0.1  | 0:0  | 248146.7        |         | 14.7             |
| Industry                                               | 341.4  | 483.6  | 2655.4 | 280.7     | 192.2             | 290.7 | 12.4            | 27.5 | 17.8 | 201241.1        | 1       | :                |
| Mobile source<br>and oil storage<br>and transportation | 18.2   | 531.2  | 785.2  | 18.8      | 16.4              | 182.8 | 11.5            | 8.9  | 2.8  | 70551.0         | 4.0     | 4.0              |
| Fugitive dust                                          |        |        |        | 441.2     | 107.5             |       | 1               | 9.0  | 8.1  |                 |         | 1                |
| Resident                                               | 26.3   | 19.3   | 775.8  | 98.5      | 79.0              | 250.0 | 7.5             | 9.1  | 48.9 | 16133.0         | 13.1    |                  |
| Agriculture                                            | 2.3    | 9.5    | 166.9  | 22.8      | 22.4              | 24.5  | 634.1           | 1.3  | 12.7 |                 | 11219.5 | 24.0             |
| Waste treatment                                        |        |        | 1      |           | 1                 | 6.3   | 4.3             | 1    | 1    | 4390.3          | 1225.4  | 0.2              |
| Total                                                  | 408.7  | 1336.2 | 4647.3 | 9011      | 440.0             | 759.3 | 672.7           | 47.4 | 90.3 | 540462.0        | 12462.0 | 42.9             |





Figure 2. Source contributions of air pollutants and GHG emissions. (a) Overall source distribution across pollutants; (b-i) Sub-category source distribution across pollutants: (b) SO<sub>2</sub>, (c) NO<sub>X</sub> (d) CO, (e) PM<sub>10</sub>, (f) PM<sub>2.5</sub>, (g) VOCs, (h) NH<sub>3</sub>, (i) CO<sub>2</sub>.

proportion of residential source emissions is attributed to the source classification methodology, which incorporates civil solvent use, architectural surface coating, asphalt paving, and pesticide use within the residential category. On the residential side, asphalt paving (9.6%) and building surface coating (6.9%) are the primary subcategories. These emissions are closely linked to large-scale urban construction, where coatings, adhesives, and asphalt materials are widely used (Liang et al., 2021). Mobile source and oil storage and transportation emerge as the predominant contributor. This substantial contribution can be attributed to Henan Province's extensive vehicle fleet, with Zhengzhou alone having 4.57 million vehicles in stock in 2022, ranking sixth nationally (HBS, 2023; HPDEE, 2022). As a province with 98.72 million permanent residents and serving as a crucial transportation hub connecting northern and southern China (HBS, 2023), Henan experiences substantial traffic volumes from both local commuting and long-distance freight transport. The province's well-developed highway network and rapid economic growth have driven increased vehicle ownership and usage, consequently elevating VOCs emissions from mobile source and oil storage and transportation.

NH<sub>3</sub>, CH<sub>4</sub>, and N<sub>2</sub>O emissions are mainly from agricultural source, accounting for 94.3%, 90.0%, and 55.8% of total emissions, respectively. Livestock farming is the main contributor to all three. Henan is a major livestock province, with 42.6 million live pigs in stock in 2022, ranking second nationally (HBS, 2023). Large amounts of manure release NH<sub>3</sub>, CH<sub>4</sub>, and N<sub>2</sub>O through volatilization, anaerobic decomposition, and denitrification, forming the key agricultural emission pathways. The province also has abundant agricultural land, with a stable arable area of 14.7 million hectares. As a major grain and






agricultural production base, Henan applies fertilizers intensively, particularly nitrogen fertilizers, which significantly contribute to increased  $NH_3$  and  $N_2O$  emissions. In terms of  $N_2O$ , 34.3% of emission is attributed to electricity and thermal power source, with nearly all (99.9%) originating from coal combustion. This highlights once again the province's dependence on coal within its energy structure.

CO<sub>2</sub> emissions are primarily from electricity and thermal power (45.9%) and industrial source (37.2%), together contributing 83.1% of total emissions. This highlights Henan's heavy dependence on coal-based energy and energy-intensive industries. Almost all emissions from the power sector (99.5%) originate from coal combustion, supported by six major coalfields and annual coal consumption exceeding 200 million tones. This coal-dominated energy structure is the primary driver of CO<sub>2</sub> emissions in the power sector (Wang et al., 2024). Industrial CO<sub>2</sub> emissions are mainly predominantly generated by high-emission sectors such as steel, chemicals, and building materials. Specifically, coal-fired industrial boilers account for 84.5% of industrial emissions, while cement production contributes 14.9%. mobile source and oil storage and transportation contribute 13.1% of total CO<sub>2</sub> emissions, with On-road mobile comprising 89.2% of this proportion. As a key transportation hub, Henan experiences intensive freight activity, particularly from heavy-duty diesel trucks, which constitute the main source of CO<sub>2</sub> emission from mobile source and oil storage and transportation (Zhang et al., 2025).

#### 3.2 Emission characteristics by city

Influenced by its industrial structure and functional positioning, the emission patterns of pollutant and GHGs emission across the study region exhibit clear geographical characteristics and synergistic variation trends (Figure 3 and Figure 4). The northern region, encompassing Anyang, Hebi, and Jiaozuo, is dominated by energy-intensive heavy industries such as steel production, non-ferrous metals processing, and building materials manufacturing. The extensive use of coal in these activities leads to high emission of SO<sub>2</sub>, NO<sub>X</sub>, and PM<sub>2.5</sub>, as well as significantly elevated levels of CO<sub>2</sub> emission intensity. Anyang exemplifies this pattern with 2022 emissions of 77259 tons SO<sub>2</sub>, 153763 tons NO<sub>X</sub>, and 59738 tons PM<sub>2.5</sub>. Its CO<sub>2</sub> emissions totaled 513.6 Mt, yielding a carbon intensity of 20444 t/10<sup>8</sup>yuan GDP-1.75 times the provincial average of 11679 t/10<sup>8</sup>yuan GDP. Source apportionment analysis indicates that industrial source contributes 95.5% of SO<sub>2</sub> and 74.6% of NO<sub>X</sub> emissions. Combined industrial source and electricity and thermal power account for over 91% of regional CO<sub>2</sub> emissions. Similar emission

profiles observed in Jiaozuo and Hebi indicate that this region as a concentrated hotspot for both air pollutants and carbon emissions.

 $Figure \ 3. \ Map \ of \ air \ pollutants \ and \ GHG \ emissions \ in \ each \ city.$ 

Figure 4. Maps of emission intensity per unit of GDP (ton/10<sup>8</sup> RMB) for (a) SO<sub>2</sub>, (b) NO<sub>X</sub>, (c) PM<sub>2.5</sub>, (d) VOCs, (e) NH<sub>3</sub>, and (f) CO<sub>2</sub> across cities. Pie charts in each map indicate the source contributions of each city.

As for the central region, with Zhengzhou as its core, serves as the province's primary hub for political, economic, and transportation. This area exhibits high NO<sub>X</sub> and VOCs emissions along with the highest CO<sub>2</sub> emission levels in the province, indicating a clear pattern of pollution dominated by motor vehicle emission. Zhengzhou recorded 125643 tons of NO<sub>X</sub> emissions in 2022, ranking second in the province, with mobile source and oil storage and transportation contributing 61.0%. The city's VOCs emissions reached 106270 tons, ranking first provincially, with mobile source and oil storage and transportation accounting for over 30% of the total. This differs significantly from the provincial emission structure dominated by industrial source, reflecting the typical emission characteristics of






megacities with dense traffic networks. Despite generating the province's largest CO<sub>2</sub> emissions (625.7 Mt), the carbon intensity remains relatively low at 4837 t/10<sup>8</sup>yuan GDP, attributed to substantial economic output representing 21.1% of provincial GDP (HBS, 2023). The city's vehicle fleet consists of 4.6 million units, accounting 23% of the provincial total, and ranks first in the province in terms of passenger vehicles, heavy-duty trucks, and new registrations (ZBS, 2023). This extensive transportation infrastructure and logistics operations make road transport the primary contributor to air pollution and carbon emissions both.

In addition, the western region, including Luoyang, Pingdingshan, and Jiyuan, utilizes its abundant coal resources to support the development of energy and chemical industries, leading to increased SO<sub>2</sub> emissions and a high carbon dioxide intensity. Jiyuan exemplifies this high-carbon emission model, with 378855 tons of SO<sub>2</sub> emission and 349.7 Mt of CO<sub>2</sub> emission in 2022. Its carbon intensity of 438378 t/10<sup>8</sup>yuan GDP is 3.7 times higher than that the provincial average. Emission source analysis reveals that industrial source contribute 94.0% of SO<sub>2</sub> emission, while industrial source (41.7%) and electricity and thermal power source (42.2%) together account for 83.9% of CO<sub>2</sub> emission.

In contrast, the southern region, encompassing Zhumadian, Zhoukou, and Xinyang, serves as a major agricultural production area characterized by larger NH<sub>3</sub> emission from farming activities but relatively lower carbon emission totals and intensities. Zhumadian illustrates this pattern with primary industry comprising 17.6% of GDP—the province's highest proportion. The substantial livestock populations, including 6.74 million pigs and 53.6 million poultry, generate 76792 tons of NH<sub>3</sub> emissions (ZBS, 2023), with agricultural source accounting for 96.5%. Conversely, regional CO<sub>2</sub> emission amount total only 163.3 million tons, resulting in an emission intensity of 5013 t/10<sup>8</sup>yuan GDP—markedly lower than that of northern industrial cities. This disparity reflects the significantly lower energy demand associated with agricultural production system compared to industrial ones.

Finally, for the eastern region, which includes Shangqiu and Xuchang, integrates industrial production, agricultural activities, and transportation functions, leading to moderate overall pollutant emissions and intermediate CO<sub>2</sub> intensities. Xuchang's emission profile exemplifies this multi-source pattern: industrial sources contribute 87% of SO<sub>2</sub> emission (22452 tons); NO<sub>X</sub> emissions (58434 tons) are predominantly from industrial source (48.9%) and mobile source and oil storage and transportation (27.2%); PM<sub>2.5</sub> emissions (20842 tons) are mainly generated by industrial (55.0%) and fugitive dust






(16.1%); NH<sub>3</sub> emissions (23510 tons) are overwhelmingly attributed to agricultural activities (94.6%) of; and VOCs emissions originate from a variety of sectors, including from industrial (52.1%), residential (24.9%), and mobile source and oil storage and transportation with petroleum operations (19.2%). This complex, multi-source emission structure adds the complexity of integrated pollution control strategies for this region.

#### 3.3 Comparison and evaluation of emission inventory

#### 3.3.1 Comparison with others

To evaluate the reliability of the emission inventory developed in this study, we conducted a comparative analysis with several existing studies and databases, as summarized in Table 2.

First, regarding GHGs, the estimated CO<sub>2</sub> emissions in this study are notably higher than those reported by MEIC and CEADs. This difference primarily stems from variations in activity data source and spatial resolution. While all studies employ the emission factor approach, our estimates are derived from individual enterprise-level accounting of 7390 designated size industrial enterprises, incorporating detailed terminal energy consumption data from power and heating sectors, using one-to-one enterprise-level accounting rather than aggregated statistical data. In contrast, MEIC and CEAD rely on provincial energy balance sheets. That being said, CHRED's CO<sub>2</sub> emissions figures for Henan Province are a tad higher than what we've come up with. We reckon this discrepancy boils down to differences in what each of us is counting as part of the whole. CHRED's throwing both direct and indirect CO<sub>2</sub> emissions into the mix, while we're just looking at the direct stuff.

In terms of air contaminants, our findings show that the projected discharges of  $SO_2$ ,  $NO_X$ ,  $PM_{10}$  and  $PM_{2.5}$  are actually more substantial than the figures provided by the MEIC. This can be attributed to the treatment of key emission source categories through enterprise-specific activity data. For industrial source, emissions were calculated through one-to-one enterprise-level accounting, particularly for high-emission industries such as cement, steel, and chemicals.

In contrast, the estimated emissions of NH<sub>3</sub>, CO and VOCs are slightly lower than those reported in MEIC. the open burning of crop residues was estimated using satellite-based fire hotspot data. The number of fire points was used to adjust burning ratios across different regions, thereby generating less region-specific emission factors. The discrepancy in CO and VOCs can be primarily attributed to the use

Table 2. Comparisons with other study.

| Region Year | Year | GDP/Billion |             |        |        |                  | Pollu             | Pollutant/kt                            |                 |                                                                     |          |        | Reference       |
|-------------|------|-------------|-------------|--------|--------|------------------|-------------------|-----------------------------------------|-----------------|---------------------------------------------------------------------|----------|--------|-----------------|
|             |      | CNY         | $SO_2$      | NOx    | 93     | PM <sub>10</sub> | PM <sub>2.5</sub> | PM <sub>10</sub> PM <sub>2.5</sub> VOCs | NH <sub>3</sub> | CO2                                                                 | CH4      | $N_2O$ |                 |
| Henan       | 2022 | 62106       | 408.7       | 1336.2 | 4647.3 | 901.1            | 440.0             | 759.3                                   | 672.7           | 408.7 1336.2 4647.3 901.1 440.0 759.3 672.7 540462.0 12461.97 42.93 | 12461.97 | 42.93  | This study      |
| Henan       | 2020 | 54691       | 232.7       | 938.0  | 5784.6 | 342.1            | 342.1 265.2       | 1181.7 789.0                            | 789.0           | 427679.2                                                            |          |        | MEIC            |
| Henan       | 2020 | 62106       |             |        |        |                  |                   |                                         |                 | 572790.0                                                            |          |        | CHRED           |
| Henan       | 2020 | 54691       |             |        |        |                  |                   |                                         |                 | 483738.0                                                            |          |        | CEADs           |
| Guangdong   | 2020 | 88521       | 181.0       | 985.0  |        |                  | 191.00            | 976.0                                   | 390.0           |                                                                     |          |        | Li et al., 2023 |
| Shanxi      | 2017 | 55216       | 131.0 381.7 | 381.7  |        | 279.5            | 279.5 737.7 206.1 | 206.1                                   |                 |                                                                     |          |        | Bo et al., 2023 |

Annotation: MEIC (Multi-resolution Emission Inventory for China, developed by Tsinghua University, http://meicmodel.org.cn/?page\_id=560); CHRED (China High-Resolution Emission Database, Developed by the Department of Earth System Science, Tsinghua University, College of Environmental Sciences and https://www.cityghg.com/toArticleDetail?id=203); CEADs (China Emission Accounts and Datasets, Developed by Institute of Climate Change and Sustainable Sciences, Environmental Jo Academy Research of Atmospheric Environment, Chinese Development, Tsinghua University, https://www.ceads.net/data/province/) Institute University; Engineering, Peking







of localized emission factors for mobile source and oil storage and transportation. These factors were adjusted using local meteorological data, vehicle age structure, and fuel standards. By employing region-specific parameters, our approach mitigates the risk of overestimation that may arise from using national average emission factors.

Additionally, to compare regional emission levels, we analyzed emission inventory data from other provinces, such as Guangdong and Shaanxi (Li et al., 2023; Bo et al., 2023). The results indicate that Henan Province generally exhibits higher emissions of most pollutants. It can be explained by differences in industrial and energy structures. As a major agricultural and industrial province in Central China, our study region relies heavily on coal and maintains intensive agricultural activities, which contribute to elevated emissions of SO<sub>2</sub>, NO<sub>X</sub>, NH<sub>3</sub>, and CH<sub>4</sub>. In contrast, Guangdong benefits from a cleaner energy structure and an economy dominated by the service sector, thereby reducing emissions. Although Shaanxi possesses a substantial industrial base, with the primary sector accounting for 7.8%, the secondary sector for 47.6%, and the tertiary sector for 44.6% (SPBS, 2023), its industrial structure differs significantly from that of Henan. Coal mining constitutes a large proportion of Shaanxi's industrial structure, with annual raw coal production consistently ranking third nationally. Additionally, Shaanxi has a lower vehicle ownership compared to Henan, resulting in emission differences between the two provinces.

# 3.3.2 Uncertainty analysis

Uncertainties in emission inventories arise mainly from the volatility and incompleteness of data on emission factor and activity date of emission source, as well as errors and under-representation of monitoring instruments. These issues are inherent challenges in constructing emission inventory (Super et al., 2020). This study employs Monte Carlo simulation to assess emission estimate uncertainties. The uncertainty ranges of emissions, simulated mean value and emission at 95% confidence intervals for each emission source and total source are presented in Table 3. Among all pollutants, the emissions of NH<sub>3</sub>, VOCs, and N<sub>2</sub>O show relatively high uncertainty. This is mainly due to the limited representativeness of emission factors and difficulties in collecting reliable activity data. For NH<sub>3</sub>, the main source of uncertainty lies in activity data, with waste treatment showing the highest uncertainty (-73%, 75%,), followed by agriculture (-54%, 52%). The primary reason is the complexity of emission processes and

Table 3. Uncertainty of the emission inventory.

| Total                                            | (-43%, 44%) | (-29%, 29%) | (-46%, 46%)  | (-51%, 48%) | (-44%, 45%)       | ) (-56%, 55%) | (-54%, 52%) | (-42%, 42%) | (-38%, 39%) | (-64%, 58%)  |
|--------------------------------------------------|-------------|-------------|--------------|-------------|-------------------|---------------|-------------|-------------|-------------|--------------|
| Waste                                            | ı           |             |              |             |                   | (-38%, 38%)   | (-73%, 75%) | (-50%, 51%) | (-60%, 64%) | (-43%, 41%)  |
| Agriculture                                      | (-33%, 34%) | (-72%, 75%) | (-105%, 99%) | (-70%,70%)  | (-73%, 67%)       | (-52%, 56%)   | (-54%, 52%) | ı           | (-28%, 30%) | (-102%, 95%) |
| Resident                                         | (-36%, 38%) | (-28%, 28%) | (-49%, 45%)  | (-58%, 55%) | (-38%, 38%)       | (-34%, 36%)   | (-31%, 32%) | (-22%, 21%) | (-40%, 39%) | 1            |
| Fugitive dust                                    | ı           | ı           | ı            | (-54%, 51%) | (-46%, 48%)       | ı             | ı           | ı           | ı           | ı            |
| Mobile source and oil storage and transportation | (-57%, 55%) | (-28%, 28%) | (-60%, 60%)  | (-42%, 2%)  | (-62%, 57%)       | (-65%, 63%)   | (-31%, 28%) | (-36%, 37%) | (-47%, 47%) | (-80%, 71%)  |
| Industry                                         | (-45%, 46%) | (-30%, 29%) | (-33%, 34%)  | (-28%, 28%) | (-44%, 45%)       | (-57%, 55 %)  | (-37%, 41%) | (-46%, 42%) | 1           | 1            |
| Electricity and thermal power                    | (-18%, 16%) | (-32%, 33%) | (-28%, 31%)  | (-38%, 40%) | (-39%, 41%)       | (-27%, 27%)   | (-37%, 34%) | (-41%, 42%) | 1           | (-39%, 38%)  |
| Category                                         | $SO_2$      | NOx         | 00           | $PM_{10}$   | PM <sub>2.5</sub> | VOCs          | $NH_3$      | $CO_2$      | $ m CH_4$   | $N_2O$       |







challenges in accurate monitoring. In waste treatment, different treatment technologies and variable waste composition contribute to emission uncertainties (Zheng et al., 2012). In agricultural activities, varying fertilization practices and changing climatic conditions significantly influence NH<sub>3</sub> volatilization, while farmers often lack detailed fertilization records (Lessmann et al., 2025). These factors collectively contribute to the substantial uncertainty in emission data. For VOCs, uncertainty mainly arises from the applicability of emission factors. In sectors such as mobile source and oil storage and transportation, emission levels vary widely due to vehicle conditions, fuel quality, and equipment aging, resulting in an uncertainty of up to -65% and 63%. For N<sub>2</sub>O, uncertainty is primarily concentrated in agricultural source. Data Gaps in data regarding livestock manure management—including treatment methods, storage duration, and application practices—make it challenging to accurately quantify emissions, particularly for small-scale farms (Hu et al., 2024).

#### 3.4 Characteristic of monthly variation and gridded spatial emission

#### 3.4.1 Temporal variation

SO<sub>2</sub>, CO, PM<sub>10</sub> and PM<sub>2.5</sub> exhibit elevated levels during winter months (January and December) (Figure 5). This is mainly due to the increase in pollutant emissions resulting from the large amount of coal and biomass combustion during the heating period in winter, while the poor atmospheric dispersion conditions in winter make it easy for pollutants to accumulate, thus leading to higher emissions. NH<sub>3</sub> and N<sub>2</sub>O, which have emission in the summer months, especially June and July, and lowest -emission in the winter months (January and December). Primarily caused by elevated NH<sub>3</sub> and N<sub>2</sub>O releases during summer months due to intensive agricultural practices, such as the application of nitrogen fertilizers and animal excreta management. In addition, high summer temperatures increase the volatilization of ammonia and soil microbial activity, further exacerbating emissions of these two pollutants. In winter, when agricultural activities are reduced and temperatures are lower, emissions of NH<sub>3</sub> and N<sub>2</sub>O are significantly lower. Emission of CH<sub>4</sub> and VOCs are relatively stable throughout the year, with small fluctuations reflecting their stable sources of emissions. The highest -emission of NO<sub>X</sub> are found in August, and the lowest emission are found in April and May. This results from higher NO<sub>X</sub> levels in summer, driven by increased vehicle emissions and intensified photochemical reactions in high temperatures.CO<sub>2</sub> is highest in December and lowest in April. Winter CO<sub>2</sub> emissions predominantly stem

from fossil fuel burning. The substantial demand for heating during winter leads to an increase in fossil fuel consumption, causing emissions to reach their peak. In contrast, the climate in April is mild; heating requirements have ceased while cooling demands have not yet commenced, resulting in energy consumption returning to baseline levels.

Figure 5. Monthly variations of air pollutants and GHG emissions in 2022 with uncertainty boundaries.

#### 3.4.2 Spatial distribution



High-emission areas of SO<sub>2</sub>, NO<sub>X</sub>, CO, PM<sub>10</sub>, PM<sub>2.5</sub>, VOCs, and CO<sub>2</sub> are mainly concentrated in cities such as Zhengzhou, Luoyang, Jiaozuo, Xinxiang, Anyang, and Hebi, along with their surrounding regions (Figure 6). These regions are the province's primary industrial hubs and population hotspots, boasting notably higher emission levels than the rest of the areas. The layout of sulfur SO<sub>2</sub> and NO<sub>X</sub> hotspots is tightly tied to the positions of coal-burning power stations, steelworks, and cement plants.

These industrial facilities are predominantly situated in heavy industrial cities like Anyang, Jiaozuo, Hebi, and Luoyang. PM<sub>10</sub> and PM<sub>2.5</sub> emissions are primarily concentrated in Zhengzhou, Luoyang, Jiaozuo, Xinxiang, and Xuchang. In the Zhengzhou metropolitan area, particulate emission exhibits a clear spatial distribution pattern, which is influenced not only influenced by industrial and fugitive dust source but



Figure 6. 3 km  $\times$  3 km gridded spatial distribution of air pollutants and GHG emissions.

490 also closely related to heavy urban traffic and high vehicle density. High-emission zones of CO and VOCs are observed in Zhengzhou and its surrounding cities, including Kaifeng, Xinxiang, and Xuchang. These cities have high road traffic density and account for more than half of the province's total vehicle ownership. In addition, the presence of petrochemical facilities, logistics centers, and fuel stations further contributes to elevated CO and VOCs emissions.

The spatial distribution of CO<sub>2</sub> is strongly correlated with energy consumption. Industrial and densely populated cities, such as Zhengzhou, Luoyang, Jiaozuo, Xinxiang, Anyang, and Hebi, exhibit relatively high levels of gridded CO<sub>2</sub> emission density. Collectively, these six cities contribute to over 51% of the province's total industrial energy consumption, amounting to more than 72 million tons of standard coal equivalent. Coal and electricity remain the dominant energy sources, leading to substantial carbon emissions.

In contrast, NH<sub>3</sub> exhibits a spatial pattern dominated by agricultural activities. Pronounced emissions concentrate across central, eastern plains with intense farming. Large-scale application of nitrogen fertilizer and livestock farming are the major emission sources of NH<sub>3</sub>. Elevated NH<sub>3</sub> levels are also observed around Zhengzhou, primarily attributed to vehicle emissions from urban traffic.

# 3.5 Identification and spatial characterization of synergistic emission hotspots

Figure 7 demonstrates the contribution of major pollutants in Henan Province in 2022 (SO<sub>2</sub>, NO<sub>X</sub>,


PM<sub>2.5</sub>, VOCs) and CO<sub>2</sub> synergistic hotspot areas contribute to the total emissions. From both pollutant and CO<sub>2</sub> emissions, co-commitment types typically contribute more to total emissions. Especially, category 1 (the grid of the top 5% of the emission intensity) occupies a large share of pollutants or CO<sub>2</sub> emissions, accounting for more than 50% of the total emissions. This indicates that the synergistic hotspots identified using gridded emission data and the top 20% quartile ranking effectively captured the spatially most concentrated emission areas with high precision. As can be seen in Figure 8, the synergistic hotspot areas of CO<sub>2</sub> and various pollutants in Henan Province exhibit significant spatial clustering characteristics, mainly concentrated in the north-central region. This concentration is particularly pronounced in the urban agglomeration centered around Zhengzhou, indicating that that this region represents overlapping zone of carbon emissions and multiple pollutant emissions, making it a critical region for synergistic management. Specifically, the synergistic hotspots of SO<sub>2</sub> and CO<sub>2</sub> are highly concentrated in the north-central region, with dense hotspot grids are observed in Zhengzhou and

Figure 7. Emission contributions (%) of five categories to different emission co-hotspot. (a) SO<sub>2</sub> and CO<sub>2</sub>, (b) NO<sub>X</sub> and CO<sub>2</sub>, (c) PM<sub>2.5</sub> and CO<sub>2</sub>, (d) VOCs and CO<sub>2</sub>. Categories represent the grad of grid based on descending order of emissions: Category 1 (top 5% grids), Category 2 (6-10%), Category 3 (11-15%),

Category 4 (16-20%), and Others (bottom 80%).

Figure 8. Maps of the co-hotspots of (a) SO<sub>2</sub> and CO<sub>2</sub>, (b) NOx and CO<sub>2</sub>, (c) PM<sub>2.5</sub> and CO<sub>2</sub>, and (d) VOCs and CO<sub>2</sub> in Henan Province.

its neighboring cities. This distribution reflects the co-driving effect of heavy industrial activities, predominantly from coal-fired power plants, iron and steel, cement, and other high-energy-consuming industries, on  $SO_2$  and carbon emissions. In contrast, the synergistic hotspots of  $NO_X$  and  $CO_2$  form a distinct belt-like distribution within the urban agglomerations, along the major transportation arteries and industrial areas. This patterns suggests that their emission sources are influenced by the combined effects of industrial combustion and motor vehicle traffic, exhibiting typical multi-source superposition characteristics. The synergistic hotspots of  $PM_{2.5}$  and  $CO_2$  demonstrate a broader distribution, in addition







to the dense hotspots in the urban core area, extending beyond the densely populated urban core to surrounding township areas. This wider dispersion indicates that their emissions are jointly influenced by a variety of sources, including industry, transportation, and residential life (e.g., heating in winter and bulk coal combustion), showcasing strong spatial diffusivity and composability. In contrast, the distribution of synergistic hotspots of VOCs and CO<sub>2</sub> is relatively dispersed. While there are still concentrated distributions in urban core areas, industrial parks, and along transportation arteries, the overall number of high-level hotspots is limited, and their spatial contiguity is weak. This indicates that the spatial coupling between VOCs emissions and CO<sub>2</sub> is relatively low, primarily originating from non-combustion industrial processes such as chemicals, petrochemicals, solvent use, and transportation.

#### **4 Conclusions**

In 2022, the emissions of major species in Henan Province were: SO2 408.7 kt (-43%, 44%), NOx at 1336.2kt (-29%, 29%), CO at 4647.3 kt (-46%, 46%), PM10 at 901.1 kt (-51%, 48%), PM2.5 at 440.0 kt (-44%, 45%), VOCs at 759.3 kt (-56%, 55%), NH3 at 672.7 kt (-54%, 52%), CO2 at 540462.0 kt (-42%, 42%), CH4 at 12462.0 kt (-38%, 39%), and N2O at 42.9 kt (-64%, 58%). with industry, electricity and thermal power, mobile source and oil storage and transportation, and agriculture as the primary emission source. Pollutants and GHGs exhibited distinct spatial distribution patterns: the northern heavy industrial region was characterized by both high carbon intensity and elevated pollution levels, with carbon intensity ranging from 1.75 to 3.7 times the provincial average; the central transportation hub was dominated by NO<sub>X</sub> and VOCs emissions; the western energy-chemical region showed prominent SO<sub>2</sub> emission; the southern agricultural region was primarily influenced by NH<sub>3</sub> emission and exhibited low carbon emission; and the eastern multifunctional region displayed a multi-source coexistence pattern.

Temporally, SO<sub>2</sub> and PM<sub>2.5</sub> peak during winter due to increased heating demand, whereas NH<sub>3</sub> and N<sub>2</sub>O emissions are higher during summer agricultural season. Spatially, high-emission areas are predominantly located in the urban agglomerations of central and northern regions, particularly in Zhengzhou, Jiaozuo, and Anyang. Synergistic hotspot analysis indicates that the top 5% high-emission grid cells account for more than 50% of total emissions, with these areas primarily concentrated in major urban agglomerations.

# 5 Implication






This study reveals pronounced spatial heterogeneity in emissions across different cities and an extremely uneven spatial distribution pattern in the study region. This pattern is consistent with observations from other provinces in the North China Plain, suggesting that regional clustering of emission hotspot is a widespread phenomenon (Wu et al., 2024; Cai et al., 2018). Furthermore, distinct emission profile variations exist among cities due to differences in economic structures: northern heavy industrial cities exhibit both high carbon intensity and elevated pollution levels, presenting substantial potential for coordinated pollution and carbon reduction; central transportation hubs are dominated by NO<sub>X</sub> and VOCs emissions, making them suitable for coordinated control strategy of mobile source; and southern agricultural regions show relatively low carbon emissions but higher NH<sub>3</sub> emission, necessitating comprehensive agricultural emission management approaches. These findings emphasize the need for region-specific, synergistic emission control strategies.

Consequently, future research and policy implementation should prioritize several key areas to promote synergistic emission reduction. First, systematically identifying high-emission hotspots can provide a solid scientific foundation for integrated air quality and climate governance. This underscores the need for enhanced collaborative emission reduction efforts that simultaneously address multiple pollutants and GHGs in these priority areas. Second, given the distinct emission patterns across regions, it is essential to develop region-specific control strategies achieve coordinated reductions of both air pollutants and GHGs. Third, high-resolution emission inventories should be leveraged to formulate more precise and coordinated management policies, thereby enhancing the overall effectiveness of synergistic emission reduction initiatives.

Data availability. The authors do not have permission to share data.

**Author contributions.** S. Yin conceived, coordinated the study and revised the manuscript. J. Li performed emission inventories, analyzed characteristics, and drafted the manuscript. C. Su provided research concepts and calculation assistance. J. Wei and M. Guan contributed to inventory development, and C. Yu provided manuscript feedback.

**Competing interests.** The contact author has declared that none of the authors has any competing interests.

**Financial support.** This work was supported by National Key R&D Program of China 590 (2024YFC3713702).

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
