# Peer review of "High-resolution Emission Inventory Development and Co-emission Hotspot Identification of Air Pollutants and Greenhouse Gases in Central Plains Region, China"

_EGUsphere, 2025_

## Referee Comment (RC1)

**General Comments:**

The study by Li et al. developed a high-resolution emission inventory of air pollutants and major greenhouse gases in a heavily polluted province of China using a bottom-up approach. The authors compiled enterprise-level activity data and emission factors from field surveys, published literature, and statistical datasets. Following validation against existing emission inventories and an uncertainty analysis, the manuscript further examined the spatial and temporal patterns of air pollutant and greenhouse gas emissions. In addition, co-hotspot regions of both types of emissions were identified, offering useful insights for future strategies aimed at mitigating air pollution and climate change. Overall, the manuscript provides valuable information for improving the understanding of emission characteristics in China. However, several important issues need to be addressed before the manuscript can be considered for publication.

**Specific Comments:**

- 1. I suggest that the authors carefully check and revise the manuscript for grammatical accuracy and appropriate word usage to improve its overall clarity and readability.
- 2. Lines 85-86: "identify high-emission grids using hotspot analysis methods": Please provide more details on the analysis methodology. Additionally, it is recommended that the authors restructure this paragraph into several sentences summarizing the main objectives of the study, highlighting the logical relationships among them, and clarifying the significance of the research.
- 3. More detailed information on the anthropogenic emission sources should be provided. For example, in addition to coal combustion, what other contributors are included in the energy structure? What agricultural products play a dominant role in this province compared with others, and what is their spatial distribution? For the seven emission categories, brief introductions of the major sub-sectors should be provided.
- 4. Methodology for emission calculation: Although the authors have provided the data sources for activity levels and emission factors, additional methodological details should be clarified in the main text:
  - (1) For the activity data, field surveys constitute a major source for many emission sectors, as shown in Table S1. Therefore, it is recommended to describe how these surveys were conducted. What were the sample sizes for different source sectors? What measures were taken to ensure that the survey data are representative and reliable?
  - (2) The values and data sources of parameters used in the equations for calculating

emission rates of specific species should be provided. In addition, the emission factors of methane (from mobile sources, oil storage and transportation, and waste treatment), nitrous oxide (from mobile sources, oil storage and transportation, agricultural sources, and waste treatment), and air pollutants from mobile sources should also be reported.

- (3) For the spatial allocation, the authors used Google Maps rather than Chinese mapping services such as Baidu Maps and Amap. Could the authors discuss the potential impacts of using different map sources on the spatial distribution of emissions?
- 5. Interpretation of the results: The authors should ensure that the results are described accurately in the main text. For example:
  - (1) In lines 242-243, Table 1 shows that the contribution of residential sources to SO2 is higher than that of the electricity and thermal power sector, which differs from the statement made by the authors.
  - (2) In Section 3.1, it should be clarified whether the percentages represent the contribution of a specific source to the total emissions of a given species, or the contribution of a sub-sector within a source category, to avoid potential confusion.
  - (3) In lines 279-281, the total contribution of firewood and crop residue combustion to  $PM_{2.5}$  is 6.77%, which differs from the 5.6% value reported in the text.
  - (4) In lines 350-351, industrial sources contribute the most to VOC emissions and have a similar contribution to CO2 as mobile sources. Therefore, it is inaccurate to state that motor vehicles are the dominant emission source in Zhengzhou.
  - (5) Lind 379: According to Figure 3, Xuchang should belong to the central region of Henan Province.
- 6. Line 337: The term "source apportionment" typically refers to analyses conducted using specific methods, such as receptor models or regional models with source apportionment techniques. Since the results here are based on emission inventories, a more accurate description would be "source contribution analysis" or "emission contribution analysis".
- 7. Evaluation of emission inventories: (1) It is recommended to indicate the version of the MEIC inventory, as well as the spatial coverage and resolution for all emission inventories, in Table 2. (2) Lines 401-403: Could the authors elaborate

on the differences between CHRED and this study, specifically clarifying what is meant by direct and indirect CO2 emissions? (3) The authors claim that the estimated VOC emissions are lower than those in the MEIC inventory, attributing this to the use of localized emission factors for mobile sources and oil storage and transportation. However, since the dominant contributors to VOC emissions are industrial and residential sources rather than mobile sources, this explanation may not fully account for the observed differences. (4) It should be noted that the reference years of the current study and the other inventories differ, which may partly explain the discrepancies in emission estimates. (5) Please provide more detailed information on the Monte Carlo simulations, including the number of simulations performed for each species from each source, the software and its version, and the model configurations used.

- 8. Lines 458-459 and 468-470: The meteorological factors mentioned only affect the concentrations of air pollutants and are unrelated to the results presented in Figure 5, which are based on the temporal allocation of annual emissions for each species.
- 9. It is suggested that the authors include a figure showing the population distribution in the supplementary materials.
- 10. Lines 507-508: What are the co-commitment types being compared against?
- 11. It is difficult to distinguish differences in the co-hotspot distributions, particularly for NOx, PM2.5, and VOCs, due to overlapping emission sources. In fact, the co-hotspot areas can be inferred directly from the spatial patterns of air pollutants and CO2 shown in Figure 6. Are there any seasonal variations in the co-hotspot areas? This information could be valuable, as the dominant air pollutants vary seasonally.
- 12. It is suggested to merge Sections 4 and 5 into a single section that summarizes the methodologies and key findings of the current study, compares them with other relevant research, and provides insights for future work.

**Technical Corrections:**

- 1. Line 37: "they are fundamentally both gaseous ....." This is not correct. Particulate matter is also an important type of pollutant emitted from anthropogenic sources.
- 2. Lines 51-55, 71-74, 396-399, and 459-463: These sentences should be rephrased

- for clarity and accuracy.
- 3. Lines 78-80: The content is repeated in the following paragraph. It is recommended to remove this sentence.
- 4. Line 89: The term "research site" is not appropriate for describing a region.
- 5. Lines 100-101: It is recommended to replace "is dominated by" with "is characterized by". Additionally, the phrase "a split of" is not appropriate in this context and should be removed.
- 6. Line 116: "By refer to..." => "By referring to"
- 7. Line 118-119: "seven emission source of" => "seven emission sources of"
- 8. Lines 130-132: It is suggested to merge these sentences with the previous paragraph.
- 9. Please ensure the consistent use of dash symbols and the correct formatting of subscripts for terminology throughout the manuscript.
- 10. In Table 1, a "-" symbol in the total PM10 emissions should be removed.
- 11. Line 250: "account for" => "accounting for"
- 12. Lines 258-259: "Same to SO2" => "Similar to SO2"; "NOx emission" => "NOx emissions"
- 13. Line 262: "industrial source" => "industrial sources"
- 14. Lines 263-265 and lines 284-286: Are those results from the current study, or are they reported in other published literature?
- 15. Line 270: ", and residentail." => ", and residential sources."
- 16. Line 272-273: "showing that" => "indicating"
- 17. Line 274: "in this region": Which region is being referred to here?
- 18. Lines 317-318: The conclusion stating a "heavy dependence on coal-based energy and energy-intensive industries" cannot be inferred solely from the contributions of these two source categories. It is recommended to remove this statement, as the subsequent analysis already addresses this point.
- 19. Title of Section 3.2: Consider revising it to "Emission characteristics at the city level".
- 20. Line 385: "of" should be removed.
- 21. Line 440: "75%,)" the comma should be removed.
- 22. It is recommended to revise the title of Section 3.3.1 to "Comparison with other emission inventories".
- 23. It is recommended to display all panels of Figure 6 on a single page.
- 24. Line 481: "Sulfur" should be removed.
- 25. Line 486: "which is influenced not only influenced by": The second "influenced" should be removed.